

# Low compositions of human toll-like receptor 7/8-stimulating RNA motifs in the MERS-CoV, SARS-CoV and SARS-CoV-2 genomes imply a substantial ability to evade human innate immunity

Chu-Wen Yang[1] and Mei-Fang Chen[2]

[1] Department of Microbiology, Center for Applied Artificial Intelligence Research, Soochow University, Taipei, Taiwan
[2] Department of Medical Research, Taipei Veterans General Hospital, Taipei, Taiwan

## ABSTRACT

**Background:** The innate immune system especially Toll-like receptor (TLR) 7/8 and the interferon pathway, constitutes an important first line of defense against single-stranded RNA viruses. However, large-scale, systematic comparisons of the TLR 7/8-stimulating potential of genomic RNAs of single-stranded RNA viruses are rare. In this study, a computational method to evaluate the human TLR 7/8-stimulating ability of single-stranded RNA virus genomes based on their human TLR 7/8-stimulating trimer compositions was used to analyze 1,002 human coronavirus genomes.

**Results:** The human TLR 7/8-stimulating potential of coronavirus genomic (positive strand) RNAs followed the order of NL63-CoV > HKU1-CoV >229E-CoV ≅ OC63-CoV > SARS-CoV-2 > MERS-CoV > SARS-CoV. These results suggest that among these coronaviruses, MERS-CoV, SARS-CoV and SARS-CoV-2 may have a higher ability to evade the human TLR 7/8-mediated innate immune response. Analysis with a logistic regression equation derived from human coronavirus data revealed that most of the 1,762 coronavirus genomic (positive strand) RNAs isolated from bats, camels, cats, civets, dogs and birds exhibited weak human TLR 7/8-stimulating potential equivalent to that of the MERS-CoV, SARS-CoV and SARS-CoV-2 genomic RNAs.

**Conclusions:** Prediction of the human TLR 7/8-stimulating potential of viral genomic RNAs may be useful for surveillance of emerging coronaviruses from nonhuman mammalian hosts.

Corresponding author
Chu-Wen Yang,
ycw6861@scu.edu.tw

## INTRODUCTION

The novel coronavirus disease 2019 (COVID-19) caused by severe acute respiratory syndrome coronavirus 2 (SARS-CoV-2) has developed into a global pandemic (*Li et al., 2020*; *Zheng, 2020*). Understanding the virology of SARS-CoV-2 and the development of
therapeutics to treat viral infection is an urgent need (*Jin et al., 2020*; *Zhou, Zhang & Qu, 2020*).

The diversity of rapidly evolved single-stranded RNA virus genome sequences among virus strains may affect viral infectivity or pathogenicity in many ways. For example, diversity in viral genome sequences may lead to changes in viral protein sequences and, consequently, changes in viral protein activity that may affect viral replication, transmission (interactions with the receptors for the virus) and antigenicity (interactions with components of the host adaptive immune system). These issues have been extensively addressed by various studies of viral protein functions (*Elfiky, 2020*; *Ou et al., 2020*; *Xia et al., 2020*). The second effect of RNA virus genome diversity is differences in protein short linear motif (SLiM) compositions among virus strains. Several viruses have been reported to interact with host cell components through viral protein SLiMs (that mimic host protein SLiMs) to facilitate manipulation of host cellular networks (*Yang, 2012*; *Via et al., 2015*). The third effect of RNA virus genome diversity is differences in genetic codon compositions. *Yang & Chen (2020)* demonstrated that the proportions of human-specific slow codons and slow di-codons in SARS-CoV and SARS-CoV-2 are lower than those in other coronaviruses. The low proportions of slow codons and slow di-codons in SARS-CoV and SARS-CoV-2 coding sequences may lead to a protein synthesis rate faster than that of other coronaviruses. The fourth effect of RNA virus genome diversity is differences in the ability to interact with components of the host innate immune system. Mammalian Toll-like receptors (TLRs) 7 and 8 are usually present in endosomal compartments, where they detect viral or endogenous single-stranded RNAs (*Streicher & Jouvenet, 2019*; *Vierbuchen, Stein & Heine, 2019*).

In the past, many studies have focused on developing TLR7 agonists to activate host inflammatory responses to cope with RNA virus infections (*Alharbi et al., 2020*; *Shah et al., 2016*). For example, TLR7 agonists were shown to provide protection against influenza A virus-induced morbidity in mice (*To et al., 2019*). TLR7 was also proposed to be a potential therapeutic target for COVID-19 (*Bonam et al., 2020*; *Poulas, Farsalinos & Zanidis, 2020*). For example, the use of imiquimod was suggested for the management of COVID-19 (*Angelopoulou et al., 2020*). On the other hand, studies have found that people with TLR7 gene variants exhibit different disease severities after SARS-CoV-2 infection (*Anastassopoulou et al., 2020*; *Lee, Lee & Kong, 2020*; *Mukherjee, Huda & Sinha Babu, 2019*). For example, epidemiological investigations found that SARS-CoV-2 affects women less severely than men (*Conti & Younes, 2020*; *Van der Made et al., 2020*). However, little is known about the potential of coronaviruses to evade detection by TLR 7/8.

Several studies have indicated that ribonucleotide compositions of RNAs are crucial for TLR 7/8 stimulation (*Forsbach et al., 2011*; *Green et al., 2012*). *Heil et al. (2004)* found that G- and U-rich single stranded RNA oligonucleotides derived from human immunodeficiency virus-1 (HIV-1) stimulate dendritic cells (DCs) and macrophages to secrete interferon and proinflammatory cytokines. Subsequently, several studies found that TLR7 recognizes G- and U-rich motifs in single stranded RNAs (*Diebold et al., 2006*;

*Forsbach et al., 2008*). Moreover, certain GU- or AU-rich RNA sequences were described to induce human TLR7- and TLR8-mediated immune responses (*Forsbach et al., 2011*; *Krüger et al., 2015*; *Zhang et al., 2018*). *Kosuge et al. (2020)* found that there is a bias to the mutations occurring in SARS-CoV-2 variants, with a preference for cytosine (C) to uracil (U) mutations. The degree of the increase in U nucleotides in SARS-CoV-2 variants correlates with enhanced production of cytokines, such as TNF-α and IL-6, in cell lines. Overall, these results indicate that genome sequence variations in RNA viruses (such as coronaviruses) may induce different degrees of human TLR7- and TLR8-mediated immune responses and, as a consequence, result in different degrees of disease severity. Therefore, genome sequence diversity may endow single-stranded RNA viruses with different abilities to evade the host TLR 7/8-mediated innate immune responses.

*Yang & Chen (2012)* developed a computational method to evaluate the human TLR 7/8-stimulating ability of single-stranded RNA virus genomes based on their human TLR 7/8-stimulating triribonucleotide compositions. In this study, the method was applied to analyze the RNA genomes of coronaviruses infecting humans. A logistic regression model was proposed for prediction of coronaviruses (from nonhuman animals) with low human TLR 7/8-stimulating activity (and, as a consequence, n higher potential to evade the human TLR 7/8-mediated innate immune responses).

## MATERIALS AND METHODS

### Data collection

The complete (full-length) genome sequences of 1,002 coronaviruses infecting humans (including 22 human coronavirus 229E (HCoV-229E), 61 human coronavirus NL63 (HCoV-NL63), 26 human coronavirus HKU1 (HCoV-HKU1), 139 human coronavirus OC43 (HCoV-OC43), 63 severe acute respiratory syndrome coronavirus (SARS-CoV), 121 Middle East respiratory syndrome coronavirus (MERS-CoV), 432 severe acute respiratory SARS-CoV-2) and 138 unclassified coronaviruses (Other) were retrieved from the Virus Pathogen Resource (ViPR, https://www.viprbrc.org/) (*Pickett et al., 2012*) and analyzed in this study (Table S1). In addition, 1,762 complete (full-length) genomic sequences of coronaviruses from six nonhuman mammalians (bat, camel, cat, civet, dog and pig) and avian hosts were retrieved from the Virus Pathogen Resource and used for analysis (Table S2).

Ninety-five oligoribonucleotides (ORNs) and 39 ribonucleotide tetramers with experimentally validated human TLR 7/8-stimulating activity were identified from 17 research reports (Table S3). The sequences of TLR 7 proteins from different organisms exhibit high variations. For example, the sequence identity of TLR 7 proteins from humans and mice is 81%. The preferences for ligand nucleotide compositions of TLRs from different organisms might be different. Since the experiments validating the TLR 7/8-stimulating activity of these ORN sequences were conducted using human cells, the TLR-stimulating triribonucleotide composition and TLR-stimulating scores described in this study should be considered to be specific for human TLR 7/8.

## Weighted triribonucleotide compositions of single-stranded RNA virus genomes

A method to evaluate the human TLR 7/8-stimulating ability of single-stranded RNA virus genomes based on their human TLR 7/8-stimulating triribonucleotide compositions was developed by *Yang & Chen (2012)*. The $4^3 = 64$ possible trimers are labeled as $X_1$, $X_2$, ..., $X_{64}$. Each trimer frequency $f_{Xi}$ is defined as

$$f_{Xi} = c_{Xi}/l, i = 1, 2, ..., 64 \tag{1}$$

here $c_{Xi}$ is the number of trimer $Xi$ and $l$ is the total number of trimers. The trimer weights $w_{Xi}$ were computed using the following formula:

$$w_{Xi} = \log_{10}\left(\frac{f_{Xi}}{1/64}\right) = \begin{cases} w_{Xi}^+ & \text{if } f_{Xi} > \frac{1}{64} \\ w_{Xi}^- & \text{if } f_{Xi} < \frac{1}{64} \end{cases} \tag{2}$$

where $w_{Xi}^+$ *and* $w_{Xi}^-$ are the weights of overrepresented and underrepresented human TLR 7/8-stimulating trimers, respectively. If the relative frequency of a trimer in the human TLR 7/8-stimulating ORN sequences is greater than 1/64 (the expected value of a random distribution), the trimer is considered to be human TLR 7/8 stimulatory. Otherwise, the trimer is considered to be nonhuman TLR 7/8-stimulatory. Each trimer is assigned a weight based on the logarithm of its relative frequency in the human TLR 7/8-stimulating ORN sequences (Fig. 1).

For any individual RNA virus genome, the positive and negative weighted trimer compositions were calculated and are referred to as Score S and Score N, respectively, these scores are collectively referred to as the human TLR 7/8-stimulating scores. Score S for stimulating trimers was calculated as

$$\text{Score } S = \Sigma(c_{Xi}w_{Xi}^+)/l \tag{3}$$

and Score N for nonstimulating trimers was calculated as

$$\text{Score } N = \Sigma(c_{Xi}w_{Xi}^-)/l \tag{4}$$

where $c_{Xi}$ is the number of trimer $Xi$ that appear in the viral genomic RNA (with $i = 1, ... 64$). $l$ is the total number of trimers in the viral genomic RNA. Higher Score S and lower Score N values indicates greater numbers of human TLR 7/8-stimulating triribonucleotides in the viral RNA genome and implies that stronger human TLR 7/8-mediated innate immunity may be induced by this viral RNA. Conversely, lower Score S and higher Score N values indicate greater numbers of human TLR 7/8 nonstimulating triribonucleotides in the viral RNA genome and, as a consequence, a higher potential for evasion of the human TLR 7/8-mediated innate immune responses (Fig. 1).

### Data analysis

Data manipulation was performed with Perl scripts written by the author. The heatmap, stripchart of Logit P values and scatter plots of Score S and Score N values were plotted

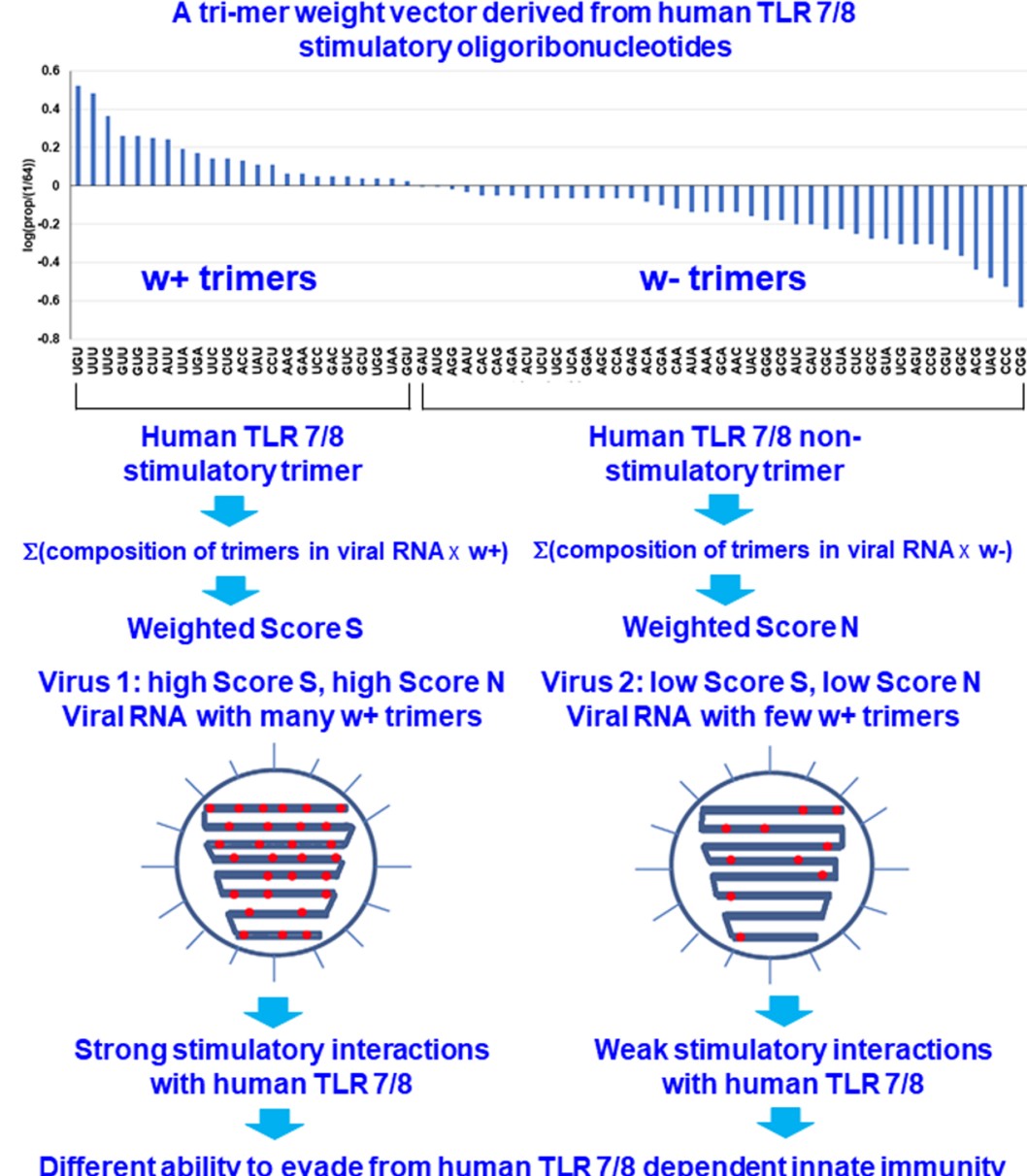

**Figure 1** Design rationale for evaluating the interactions between viral RNAs and human toll-like receptors 7/8 by analyzing the human TLR 7/8 stimulatory triribonucleotide composition of viral RNAs.                                         

using the ggplot2 package of R (the R package for statistical computing). Logistic regression was conducted with the glm function of R. Linear discriminant analysis (LDA) and quadratic discriminant analysis (QDA) were performed with the lda and qda functions, respectively, in the MASS package of R. Naive Bayes and support vector machine (svmLinear, svmPoly and svmRadial) classifiers in the caret package of R were used.

## Tenfold cross-validation

The cross-validation test is well-known and has been used in many computation-based studies (*Le, Ho & Ou, 2017*; *Le et al., 2019*; *Le, Yapp & Yeh, 2019*). The weighted triribonucleotide compositions of coronavirus genomes were randomly split into 10 subsets. Ten pairs of training (90% data for model building) and test (10% data for model evaluation) data were combined from the 10 subsets and used to perform 10-fold cross validations. For each test run, a training set was used to train a model, and the model was then tested using the test set. All methods (logistic regression, LDA, QDA, naive Bayes, svmLinear, svmPoly and svmRadial) were used to perform 10-fold cross validations. Sensitivity was computed with the following formula: sensitivity = true positive/(true positive + false negative). Specificity was computed with the following formula: specificity = true negative/(true negative + false positive). Accuracy was computed with the following formula: accuracy = (true positive + true negative)/total number of samples.

## Analysis of coronaviruses from nonhuman animals

Seven models derived from the seven methods (logistic regression, LDA, QDA, naive Bayes, svmLinear, svmPoly and svmRadial) were used to analyze data of coronaviruses from nonhuman animals. All seven methods were selected for binary classification to distinguish human coronaviruses causing common colds and severe acute respiratory syndromes. Using the results of logistic regression as a standard, the overall agreements between the results of the logistic regression model and those of the other 6 models were computed. The overall agreements were computed by the following formula: (true positive + true negative)/total number of samples.

# RESULTS

## Compositions of triribonucleotides in genomes of coronaviruses infecting humans

The similarity of triribonucleotide compositions was not consistent with the similarity of genome sequences (phylogenetic analysis in Fig. 2A). The overall triribonucleotide compositions of genomic (plus) strand and complementary (minus) strand RNAs of coronaviruses infecting humans are shown in Fig. 2B. These results indicate that triribonucleotide compositions provide novel information that cannot be revealed by phylogenetic analysis.

## Human TLR 7/8-stimulating potential of human coronavirus genomes

The human TLR 7/8-stimulating scores of coronavirus genomic (positive strand) RNAs are shown in Fig. 3. The human TLR 7/8-stimulating potential of coronavirus genomic (positive strand) RNAs followed the order of NL63-CoV > HKU1-CoV >229E-CoV $\cong$ OC63-CoV > SARS-CoV-2 > MERS-CoV > SARS-CoV. The human TLR 7/8-stimulating scores of the complementary (negative) strand of coronavirus genomic RNAs (replication intermediates) are shown in Fig. 4. The human TLR 7/8-stimulating potential of the complementary (negative) strand of coronavirus genomic RNAs followed the order

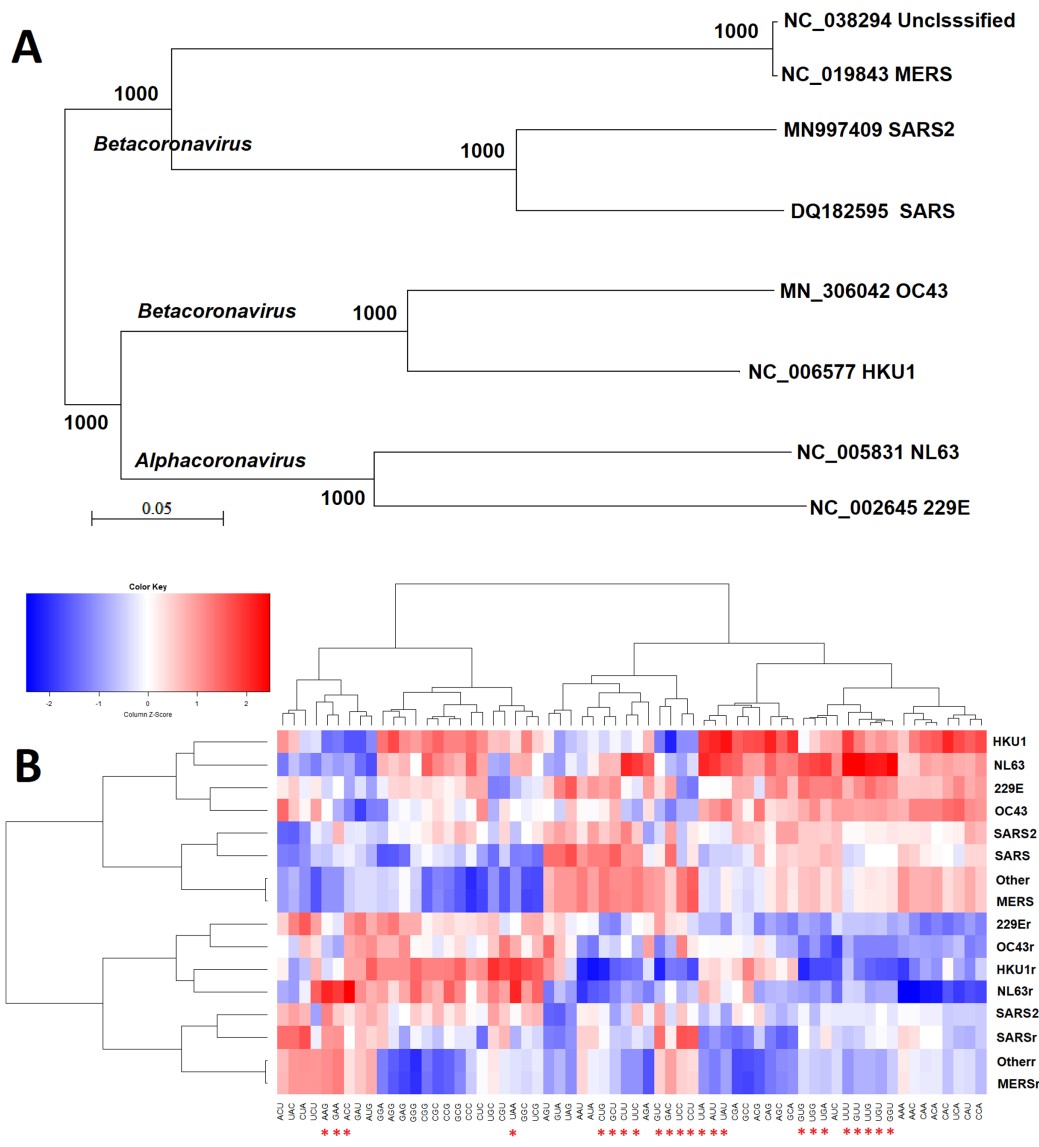

**Figure 2 Comparison of sequence and ribonucleotide trimer compositions of coronaviruses infect human hosts.** Comparison of analysis results from the conventional phylogenetic method and the weight triribonucleotide compositions proposed in this study (A) Phylogenetic analysis of full-length genomic sequences of representative coronaviruses infecting human hosts. The number of each branch obtained from 1,000 bootstrapping is shown. The accession numbers indicate the representative genomic sequences used in this analysis. (B) Comparison of triribonucleotide compositions of coronaviruses infecting human hosts. 229E: human coronavirus 229E. NL63: human coronavirus NL63. HKU1: human coronavirus HKU1. OC43: human coronavirus OC43. SARS: severe acute respiratory syndrome coronavirus (SARS-CoV), MERS: Middle East respiratory syndrome coronavirus (MERS-CoV). SARS2: severe acute respiratory syndrome coronavirus 2 (SARS-CoV-2). Other: unclassified coronaviruses. The lowercase r notations indicate the complementary (−) strands of genomic RNAs (replication intermediates). The red stars (*) indicate human TLR 7/8 stimulatory triribonucleotides.

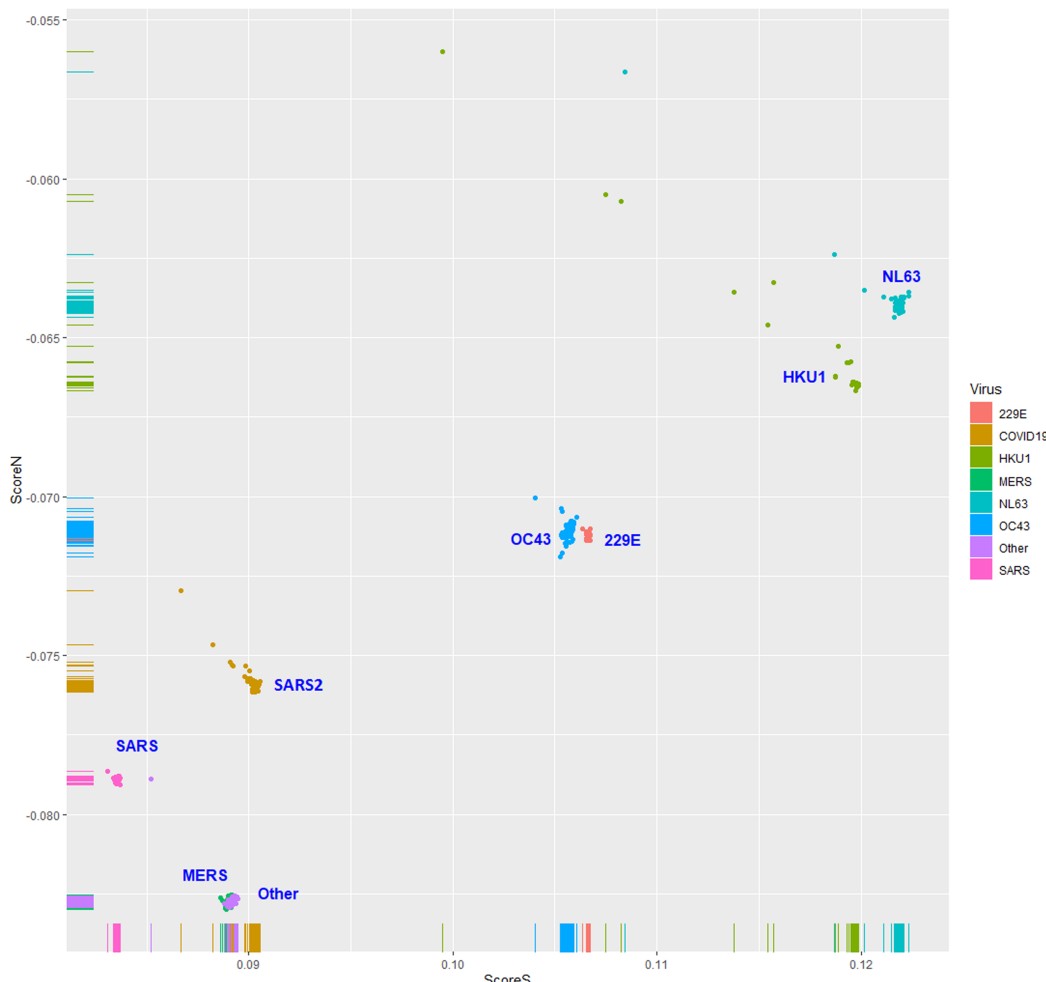

**Figure 3 The human toll-like receptor 7/8-stimulating scores of coronavirus genome (positive strand) RNAs.** Human TLR 7/8-stimulating scores of coronavirus genomic (positive strand) RNAs. Score S: human TLR 7/8 stimulating score. Score N: human TLR 7/8 non-stimulating score. 229E: human coronavirus 229E. NL63: human coronavirus NL63. HKU1: human coronavirus HKU1. OC43: human coronavirus OC43. SARS: severe acute respiratory syndrome coronavirus (SARS-CoV), MERS: Middle East respiratory syndrome (MERS-CoV). SARS2: severe acute respiratory syndrome coronavirus 2 (SARS-CoV-2).               

of SARS-CoV-2 > SARS-CoV >229E-CoV > MERS-CoV > NL63-CoV ≅ OC63-CoV > HKU1-CoV. The highest Score S value of the complementary (negative) strand of SARS-CoV-2 genomic RNA wass 0.0813 which was less than the minimum Score S value of SARS-CoV genomic (positive strand) RNA. Greater numbers of human TLR 7/8-stimulating triribonucleotides were found in the genomic (positive strand) RNAs than in the complementary strand RNAs of all coronaviruses analyzed. These results suggest a greater potential for human TLR 7/8-stimulating activity in the early stage of viral infection than the complementary (negative) strand RNAs produced during coronavirus replication stage.
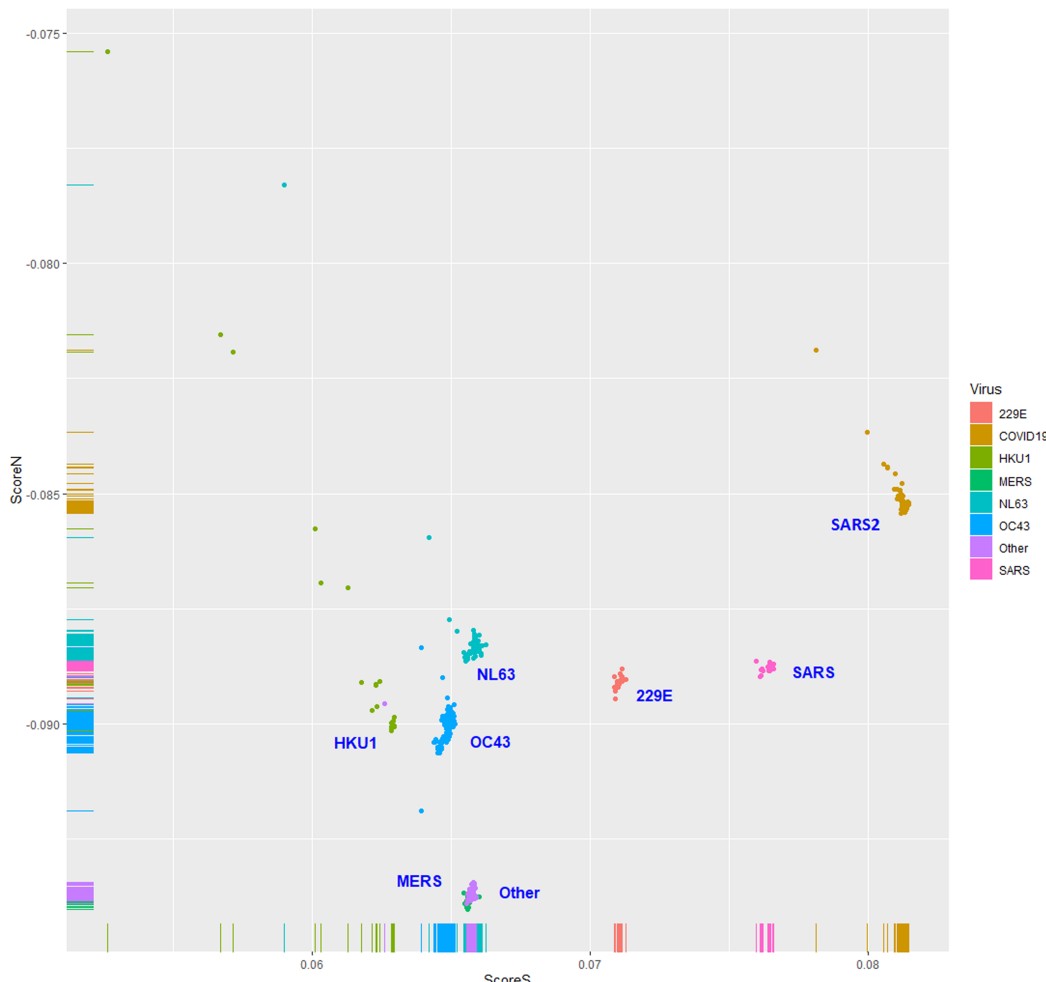

**Figure 4 The human toll-like receptor 7/8-stimulating scores of the complementary (negative) strand of coronavirus genomic RNAs (replication intermediates).** Human TLR 7/8-stimulating scores of the complementary (negative) strands of coronavirus genomic RNAs (replication intermediates). Score S: human TLR 7/8-stimulating score. Score N: human TLR 7/8 non-stimulating scores 229E: human coronavirus 229E. NL63: human coronavirus NL63. HKU1: human coronavirus HKU1. OC43: human coronavirus OC43. SARS: severe acute respiratory syndrome coronavirus (SARS-CoV), MERS: Middle East respiratory syndrome (MERS-CoV). SARS2: severe acute respiratory syndrome coronavirus 2 (SARS2-CoV-2). 

## A logistic regression model for predicting the human TLR 7/8-stimulating potential

To predict the human TLR 7/8-stimulating potential of coronaviruses, a logistic regression model was constructed as follows:

$$\text{logit}(p) = \log(p/1 - p) = \beta_0 + \beta_1 \cdot S + \beta_2 \cdot N + \beta_3 \cdot Sr + \beta_4 \cdot Nr \tag{5}$$

$S$ and $N$ are the Score $S$ and Score $N$ values for the positive strand of a viral genome sequence. Sr and Nr are the Score $S$ and Score N values for the negative strand of a viral genome sequence. 229E-CoV, OC43-CoV, NL63-CoV and HKU1-CoV were used as the highly stimulating group. MERS-CoV, SARS-CoV and SARS-CoV-2 were used as the

**Table 1 Coefficients of logistic regression from the 10-fold cross-validation and from all data.**

|        | Intercept ($\beta_0$) | $p$-Value | S ($\beta_1$) | $p$-Value | Sensitivity | Specificity | Accuracy |
|--------|----------|---------|---------|---------|-------------|-------------|----------|
| Set01  | −69.14   | 7.48e-13 | 711.33 | 1.10e-11 | 1 | 1    | 1    |
| Set02  | −78.48   | 1.92e-07 | 809.73 | 8.88e-07 | 1 | 0.96 | 0.99 |
| Set03  | −69.12   | 7.92e-13 | 711.08 | 1.17e-11 | 1 | 1    | 1    |
| Set04  | −69.05   | 8.11e-13 | 710.35 | 1.20e-11 | 1 | 1    | 1    |
| Set05  | −69.05   | 8.11e-13 | 710.35 | 1.20e-11 | 1 | 1    | 1    |
| Set06  | −69.15   | 7.56e-13 | 711.45 | 1.12e-11 | 1 | 1    | 1    |
| Set07  | −69.16   | 7.47e-13 | 711.58 | 1.10e-11 | 1 | 0.96 | 0.99 |
| Set08  | −67.95   | 1.07e-13 | 697.92 | 1.96e-12 | 1 | 1    | 1    |
| Set09  | −69.16   | 7.50e-13 | 711.56 | 1.11e-11 | 1 | 1    | 1    |
| Set10  | −79.87   | 4.31e-09 | 819.50 | 2.22e-08 | 1 | 0.96 | 0.99 |
| Mean   | −71.01   |          | 730.49 |          |   |      |      |
| Std    | 4.32     |          | 44.59  |          |   |      |      |
| All    | −69.05   | 8.11e-13 | 710.36 | 1.20e-11 |   |      |      |

Note:
Set01–Set10 using 9/10 data (as training data) was used to do regression, and 1/10 data as test data. All: 10/10 data was used to do regression.

poorly stimulating group. After the model selection procedure, the following model was selected as the final model:

$$\mathrm{logit}(p) = \log(p/1 - p) = \beta_0 + \beta_1 \cdot S \qquad (6)$$

The results of 10-fold cross-validation are shown in Table 1. The averages of the intercepts and coefficients of the 10 models were used to construct the logistic regression model as follows:

$$\mathrm{logit}(p) = \log(p/1 - p) = -71.01 + 730.49 \cdot S \qquad (7)$$

The logistic regression model using all data is:

$$\mathrm{logit}(p) = \log(p/1 - p) = -69.05 + 710.36 \cdot S \qquad (8)$$

## Comparison with other methods

Six methods were used to validate the results of logistic regression. The results of 10-fold cross-validations using linear discriminant analysis (LDA), quadratic discriminant analysis (QDA), naive Bayes and three support vector machine classifiers (linear, polynomial and radial) were the same as those using the logistic regression model (Table 1). The high values ($\cong 1$) of sensitivity, specificity and accuracy may be due to the almost complete separation of the Score $S$ values of the two groups. These results are consistent with the data shown in Fig. 3.

## DISCUSSION

The innate immune system especially the TLR 7/8-interferon pathway constitutes an important first line of defense against single-stranded RNA viruses

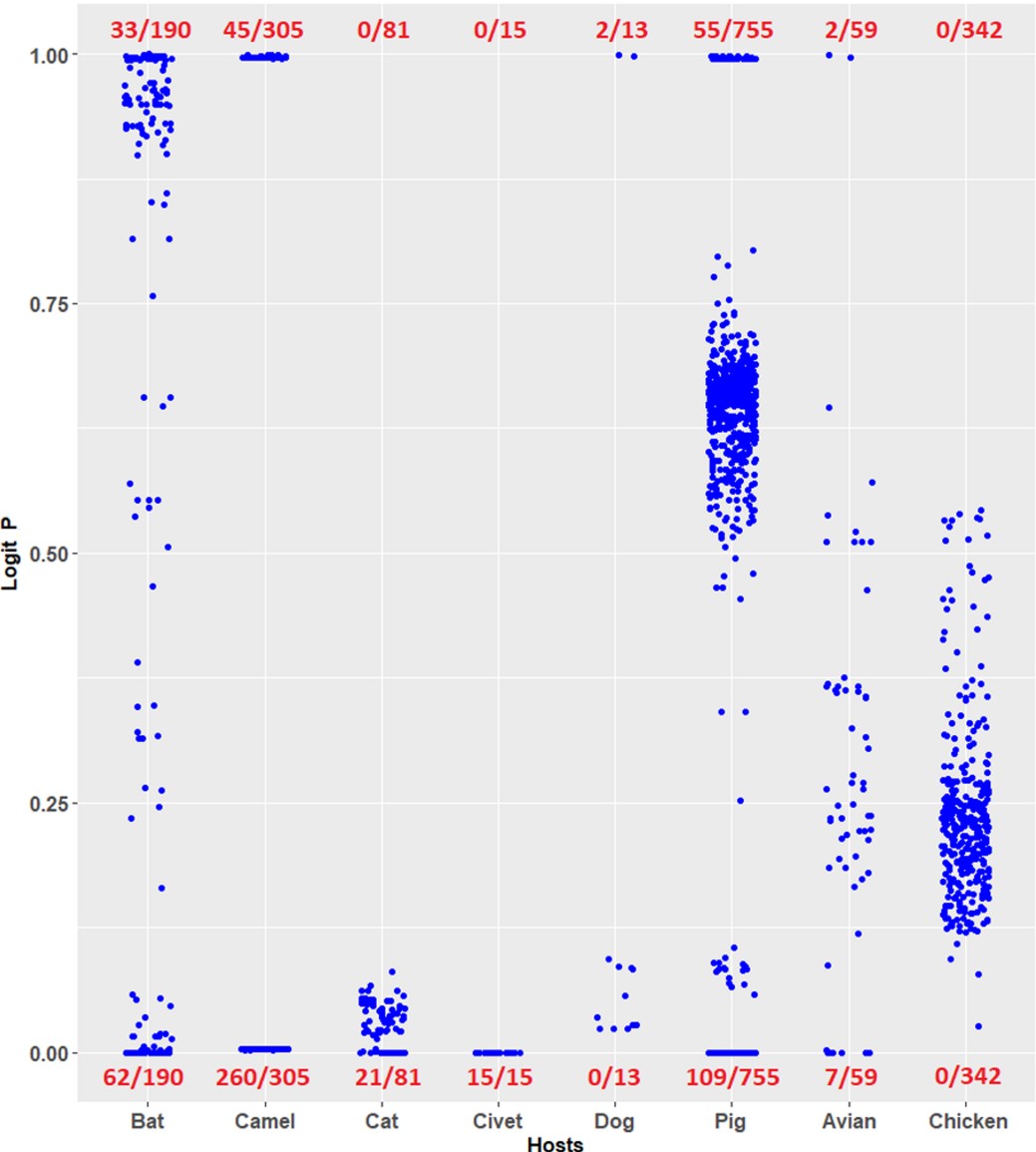

**Figure 5 Prediction of human toll-like receptor 7/8-stimulating ability of coronaviruses from mammalian and avian hosts by the logistic regression model constructed from human coronavirus data.** Prediction of human TLR 7/8-stimulating ability of coronaviruses from mammalian and avian hosts by the logistic regression model constructed from human coronavirus data. P ≅ 1 indicates a human TLR 7/8-stimulating ability equivalent to that of 229E, NL63, OC43 and HKU1 genome RNAs. P ≅ 0 indicates a human TLR 7/8-stimulating ability equivalent to that of MERS-CoV, SARS-CoV and SARS-CoV-2 genomic RNAs. The number of viruses analyzed is shown in the denominators. Avian: birds other than chickens.

(*Vierbuchen, Stein & Heine, 2019*; *Kikkert, 2020*; *Nelemans & Kikkert, 2019*). Conversely, RNA viruses have evolved multiple strategies to evade host innate immune responses to increase the success rate of infection. Several molecular mechanisms by which positive-sense single-stranded RNA viruses evade innate immune responses have been identified (*Ye et al., 2020*). Since interferons have great protective effects during early viral infection, evasion of the immune response may have differential effects on the clinical

**Table 2 Overall agreements of the logistic regression model compared with the other six models.**

|         | lda  | qda  | nb   | svmL | svmP | svmR |
|---------|------|------|------|------|------|------|
| Avian   | 0.86 | 0.34 | 0.86 | 0.88 | 0.88 | 0.17 |
| Bat     | 0.86 | 0.93 | 0.88 | 0.96 | 0.96 | 0.65 |
| Camel   | 1    | 1    | 1    | 1    | 1    | 0.96 |
| Cat     | 1    | 0.81 | 0.95 | 1    | 1    | 0.01 |
| Chicken | 0.97 | 0.35 | 0.97 | 0.97 | 0.97 | 0.03 |
| Civet   | 1    | 1    | 1    | 1    | 1    | 0.94 |
| Dog     | 1    | 1    | 1    | 1    | 1    | 0.15 |
| pig     | 0.25 | 0.84 | 0.26 | 0.81 | 0.78 | 0.82 |

**Note:**
lda, linear discriminant analysis; qda, quadratic discriminant analysis; nb, naive bayes; svmL, SVM linear classifier; svmP, SVM polynomial classifier; svmR, SVM radial classifier.

outcome of viral disease (*De Marcken et al., 2019*). Therefore, prediction of the human TLR 7/8-stimulating activities of genomic RNAs of single-stranded RNA viruses provides a method to evaluate the risk potential posed by emerging single-stranded RNA viruses.

The results of this study suggest that the genomic (positive strand) RNAs of MERS-CoV, SARS-CoV and SARS-CoV-2 are composed of low proportions of human TLR 7/8-stimulating triribonucleotides. The weak human TLR 7/8-stimulating potential of the genomic (positive strand) RNAs of MERS-CoV, SARS-CoV and SARS-CoV-2 may lead to a high ability to evade the human TLR 7/8-mediated innate immune responses during the initial stage of viral infection. In contrast, the strong human TLR 7/8-stimulating potential of the genomic (positive strand) RNAs of 229E-CoV, NL63-CoV, OC43-CoV and HKU1-CoV may confer a high probability of triggering strong human TLR 7/8-mediated innate immune responses. Different strengths of TLR 7/8-mediated innate immune responses during the initial stage of viral infection may lead to different clinical outcomes of the disease (*Frieman & Baric, 2008*; *Wong, Lui & Jin, 2016*; *Yokota, Okabayashi & Fujii, 2010*). Evaluation of the human TLR 7/8-stimulating potential of viral genomic RNAs may be useful for surveillance of emerging coronaviruses.

Coronavirus infections have been considered novel emerging zoonotic diseases (*Streicher & Jouvenet, 2019*; *Salata et al., 2019*; *Menachery et al., 2020*). Evaluating the risk potential posed by zoonotic coronaviruses is necessary. The logistic regression model constructed in this study can be used to evaluate the human TLR 7/8-stimulating potential of genomic RNAs of coronaviruses from other mammals and birds. For example, the human TLR 7/8-stimulating potential of 1,361 coronavirus genomic (positive strand) RNAs from six mammalian (bat, camel, cat, civet, dog and pig) and avian hosts were computed using the logistic regression model (Eq. 7). Logit $P \cong 1$ indicates a human TLR 7/8-stimulating ability equivalent to that of general human coronavirus (229E-CoV, NL63-CoV, OC43-CoV and HKU1-CoV) genomic RNAs. Logit $P \cong 0$ indicates weak human TLR 7/8-stimulating potential equivalent to that of the highly pathogenic coronavirus (MERS-CoV, SARS-CoV and SARS-CoV-2) genomic RNAs. As shown in Fig. 5, many of the coronavirus genomic (positive strand) RNAs from bats, camels, cats, civets and pigs exhibit weak human TLR 7/8-stimulating potential equivalent to that of

highly pathogenic coronavirus (MERS-CoV, SARS-CoV and SARS-CoV-2) genomic RNAs. Six methods were used for comparison with the logistic regression model. Using the results of logistic regression as a standard, the overall agreements between results of logistic regression model and those of the other six models are shown in Table 2. Most of the analysis results (except the results from the SVM radial classifier) were consistent with the prediction of the logistic regression model. The predictions obtained using the logistic regression model proposed in this study suggest that the routes and risks of contact with those animals (the natural reservoirs of animal coronaviruses) should be addressed. This observation may be an important key point to prevent the outbreak of emerging infectious diseases.

## CONCLUSIONS

The results of this study suggest that MERS-CoV, SARS-CoV and SARS-CoV-2 may have a relatively low human TLR 7/8-stimulating potential and relatively high ability to evade human TLR 7/8-mediated innate immune responses. Prediction of the human TLR 7/8-stimulating potential of viral genomic RNAs may be useful for surveillance of emerging coronaviruses from nonhuman animal hosts.

### Funding
The authors received no funding for this work.

### Competing Interests
The authors declare that they have no competing interests.

### Author Contributions
- Chu-Wen Yang conceived and designed the experiments, performed the experiments, analyzed the data, prepared figures and/or tables, authored or reviewed drafts of the paper, and approved the final draft.
- Mei-Fang Chen analyzed the data, prepared figures and/or tables, and approved the final draft.

### Data Availability
All sequences used in this study were retrieved from the Virus Pathogen Resource (ViPR, https://www.viprbrc.org/). Accession numbers are available in the Supplemental Files.

### Supplemental Information
Supplemental information for this article can be found online at http://dx.doi.org/10.7717/peerj.11008#supplemental-information.

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
