# Peer review of "Low compositions of human toll-like receptor 7/8-stimulating RNA motifs in the MERS-CoV, SARS-CoV and SARS-CoV-2 genomes imply a substantial ability to evade human innate immunity"

_PeerJ, doi:10.7717/peerj.11008_

## Round 0.1 · original submission · Major Revisions

Please address all the concerns of the reviewers, in particular, try to improve the language, if possible seeking help from native speakers, widen the literature cited, compare with other methods and, most important, comment on the almost perfect performances of your analysis, which might cast doubt on how the method was evaluated.

Reviewer 1 ·

Basic reporting

English language should be improved. There are some grammatical errors and typos in this manuscript. The authors should re-check and revise carefully.

Literature review is weak. It is necessary to provide more related references for genomic analysis.

Research question is not much novelty. It is like a narrow study on SAR-COV since there are a lot of works that had been done on general viruses. The users/readers could even use the general methods for this analysis.

Experimental design

Source codes should be released for replicating the methods and results.

The authors reported the measurement metrics i.e. sensitivity, specificity, or accuracy, but I did not see any mention in methodology.

Cross-validation testing is well-known and has been used in previous genomics-based studies such as PMID: 28643394, PMID: 31277574, and PMID: 31921391. Therefore, it is necessary to refer to more works in this description.

Validity of the findings

The performance results look perfect (sensitivity, specificity, accuracy of 100%). It is a point that needs to be discussed.

The authors did not compare the performance results among different methods.

It is important to compare the performance results with the previously published works in this field.

Additional comments

No comment.

Reviewer 2 ·

Basic reporting

This article is of very good help for immunology experts who were working on wet lab side. Bioinformatic analysis most of the times opens the prerequisites of where to start. We know like Covid-19 is on one hand acute at present and there fore agressive immune reactions from systemic immunity were creating cytokine storm. This cytokine storm is becoming huge problem.
In all these persuits insilico analysis is of significant help for even vaccine development work flow.

Experimental design

experimental design is pretty straight forward but might have been compared with other existing methodology as well.

Validity of the findings

Findings were of significant importantance and were valid.

Additional comments

Yes your article and analysis is very straight forward, I appreciate it. But might have validated data with more than one method.

---

## Round 0.2 · accepted · Accept

I think that the subject of how SARS-COV-2 interacts with human innate immunity is an important one, and this work can yield useful insights. The reviewers have been satisfied with your revised version, but the second reviewer would have liked that you address mechanistic aspects, which can be a useful suggestion for the follow-up of your work.

Reviewer 1 ·

Basic reporting

no comment

Experimental design

no comment

Validity of the findings

no comment

Additional comments

My previous comments have been addressed satisfactorily.

Reviewer 2 ·

Basic reporting

Authors addressed the important aspect of predicting the potential simulator affect of viral genomes on human toll like receptors 7/8. they compared such stimulatory affects among various pathogenic SARS viruses but not with focus on mechanistic variations among them.
host innate immunity mostly serves as static entity and based on this constant variable, comparing and computing the method to validate stimulatory and evasion affects on innate immune response is however appreciated.

Experimental design

The basic methodology looks good but computing on raw sequences holds less promising as authors ignored mechanistic variables.

Validity of the findings

Validity of findings with in the methods followed is fine, that helps people use the approach given by authors. But added more variables would have been better.